# Analysis of Protection and Immune Response against *Teladorsagia circumcincta* in Goats Immunised with Thiol-Binding Proteins from Adult Worms

**DOI:** 10.3390/vaccines12040437

**Published:** 2024-04-18

**Authors:** Leire Ortega, Jessica Quesada, Antonio Ruiz, Magnolia María Conde-Felipe, Otilia Ferrer, María del Carmen Muñoz, José Adrián Molina, Francisco Rodríguez, José Manuel Molina

**Affiliations:** 1Parasitology Unit, Department of Animal Pathology, Faculty of Veterinary Medicine, University of Las Palmas de Gran Canaria, 35413 Arucas, Spain; leireortegar@gmail.com (L.O.); jesesita@hotmail.com (J.Q.); otilia.ferrer@ulpgc.es (O.F.); mariadelcarmen.munoz@ulpgc.es (M.d.C.M.); jose.molina108@alu.ulpgc.es (J.A.M.); josemanuel.molina@ulpgc.es (J.M.M.); 2Clinical Veterinary Hospital, Faculty of Veterinary Medicine, University of Las Palmas de Gran Canaria, 35413 Arucas, Spain; 3Department of Anatomy and Compared Anatomy Pathology, Faculty of Veterinary Medicine, University of Las Palmas de Gran Canaria, 35413 Arucas, Spain; francisco.guisado@ulpgc.es

**Keywords:** *Telacorsagia circumcincta*, thiol-binding proteins, immunisation, immune response, goat

## Abstract

In view of the increasing occurrence of anthelmintic-resistant strains of gastrointestinal nematodes in ruminants, various alternative control strategies have been investigated, such as those based on the induction of protective immune responses by immunisation with parasite antigens. In this study, the protective activity of somatic antigens from adult worms of *Teladorsagia circumcincta* purified by affinity chromatography on thiol-sepharose was analysed in goats. After challenge, the enriched products induced a slight reduction in the cumulative faecal egg counts (21%) and in the number of worms (23.3%), with a greater effect on female worms, which also showed a reduction in parameters related to their fertility. These parasitological findings were associated with a Th2 immune response, with a prominent local humoral response and an eosinophilic infiltrate in the gastric mucosa (negatively associated with the fertility of female worms and the number of worms, respectively), as well as an infiltration of MCHII+, CD4+, IgG+ and IgA+ cells. However, several analyses showed an increase in CD8+ cells in the mucosa, as well as IL-2 expression in the gastric lymph nodes, which may have been associated with inhibition of protective responses or with the development of mixed Th1/Th2 responses, a finding that should be analysed in future studies.

## 1. Introduction

Gastrointestinal nematodes (GINs) are a major cause of economic losses in ruminants through the reduction in productivity in meat and milk production. Additionally, they contribute to reproductive disorders and have a detrimental impact on the quality of ruminant products, thereby compromising animal welfare. Among the different species that can affect the gastrointestinal tract of small ruminants, *Teladorsagia* (*T*.) *circumcincta* is particularly relevant in goats due to its wide geographic distribution [1,2] and its capacity to trigger the aforementioned negative effects, even in subclinical infections [3].

The control of these infections is mainly based on the strategic use of anthelmintics. The increasing emergence of strain resistance of these drugs, especially in goats, because of the off-label use of these compounds in this species, among other factors [4,5], and the presence of pharmacological residues in food products [6], have generated the possibility of using alternative measures, such as those based on the induction of protective immune responses, as a sustainable component of integrated parasite control programs [7]. Despite the growing interest in developing vaccines against GINs, limited research has been conducted in goats compared with other ruminant species, such as sheep [8,9]. Specifically, in the case of *T. circumcincta* vaccines, no previous trials have been undertaken in goats, although a prototype vaccine has already been developed for sheep, based on the use of various recombinant proteins of this parasite [10]. Similarly, immunoprotection trials against bovine-adapted species within the genus *Ostertagia* (*O.*) *ostertagi* have been previously conducted [11,12].

It is generally accepted that the response to GIN infections in sheep or cattle cannot be extrapolated to goats due to factors such as delayed development of immune responses in goats—that makes them less effective [13]—or the genetic differences in the immune response against GINs among goat breeds [14]. These factors, among others, underscore the importance of conducting specific studies on immunisation against GINs in goats.

Since 2004, our research group has carried out several vaccination trials against *Haemonchus* (*H.*) *contortus* in goats. In these studies, in which somatic products of adult parasite worms purified by thiol-sepharosa affinity chromatography were used as immunogen (TBSP), it was observed that the antigens present in the purified extract effectively induced a protective response in vaccinated animals, leading to a reduction in both the number of worms and the faecal egg counts [15,16,17]. These protective responses have been characterised as Th-2 and are associated with both local and systemic humoral responses elicited against antigens that may be excreted/secreted by the parasite during the development of parasitic larval stages, as well as by adult worms [18,19].

Therefore, the aim of this study was to evaluate the immunoprotective activity elicited in goats by somatic antigens of *T. circumcincta* adult worms soluble in PBS, Tween 20 and Triton X-100, and purified by a similar method to that previously used against *H. contortus.* The immune response induced by this immunogen was also analysed using protocols similar to those used in vaccination trials with TBSP fractions from *H. contortus* adult worms.

## 2. Materials and Methods

### 2.1. Preparation of Homogenates of Adult Worms of T. circumcincta and Enriched Protein Fractions by Thiol-Sepharose Chromatography

Homogenates of adult *T. circumcincta* worms were obtained as previously described [19], with some modifications. Adult worms were harvested from the abomasum of goats inoculated with L3 of a *T. circumcincta* strain kindly provided by Dr. Uriarte (Centre for Agricultural Research and Technology of Aragón (CITA), Zaragoza, Spain). Briefly, adult worms were homogenised on ice in PBS at the rate of 300 worms/mL, centrifuged and the supernatant was filtered (0.22 μm). This homogenate was referred to as S1. The sediment, from the previous centrifugation, was kept cold in 3 mL of phosphate-buffered saline (PBS) with 0.1% (*v*/*v*) Tween 20 (polyoxyethylene-sorbitan monolaurate) (Sigma-Aldrich, St. Louis, MO, USA). After further homogenisation, the sample was centrifuged, and the supernatant was filtered. This extract was referred to as S2. Finally, this second pellet was resuspended with 5 mL of PBS containing 2% (*v*/*v*) reduced Triton-X-100 (polyoxyethylene (10) isooctylcyclohexyl ether) (Sigma-Aldrich) and kept for 1 h at 4 °C under agitation. Subsequently, it was centrifuged and filtered again. The final solution was referred to as S3 [20]. Homogenates S1, S2 and S3 were mixed proportionally, and total protein concentration was quantified via BCA protein assay (Pierce^®^ BCA Protein Assay Kit, Thermo Scientific, Waltham, MA, USA) following the manufacturer’s instructions.

Extracts previously incubated with DTT (dithiothreitol) (Sigma-Aldrich) at a final concentration of 2.5 mM for 30 min at 37 °C were desalted using HiTrap desalting columns (Amersham Pharmacia Biotech, Upsala, Sweden) and then applied to a thiol-sepharose column (Amersham Pharmacia Biotech, Upsala, Sweden). The bound material was eluted with PBS buffer containing 50 mM DTT. The proteins present in these elutions, from which DTT was removed by chromatography on a HiTrap desalting column, were termed thiol-sepharose-binding proteins (TSBPs) [21]. The protein extract obtained is shown in Figure 1.

### 2.2. Experimental Design

Fifteen Majorera Canaria kids were used in this study. On arrival at the experimental facilities, the animals were treated with fenbendazole (Panacur^®^ 10% MSD-Animal Health) and kept in nematode-free conditions until the start of the experiment three months later. Goats were randomly divided into groups. Group 1 (immunised group) (*n* = 5) contained goats immunised with TSBPs and monospecifically infected with *T. circumcincta* L3. Group 2 (control group) (*n* = 5) contained non-immunised goats monospecifically infected with *T. circumcincta* L3. For comparative purposes, data obtained from experimental group 3 (*n* = 5) (non-immunised and non-infected goats) were used to analyse some parameters (relative cytokine expression).

In group 1, TSBPs were inoculated intramuscularly in complete (first inoculation, 100 µg TBSP, week 0) and incomplete (second inoculation, 100 µg TBSP, week 2) Freund’s adjuvant. In group 2 (vaccination control), TSPB was replaced by elution buffer.

Two weeks after the second immunisation (week 4), animals were challenged with 8000 infective *T. circumcincta* L3 and, at week 10 of the experiment, all animals were sacrificed. Blood samples were collected weekly for biopathological and serological analyses, At the end of the study, worm counts were performed and gastric and abomasal lymph node samples were collected for histopathological and relative cytokine expression analyses [18,19]. The experiments were conducted in accordance with European Community Council Directive 2010/63/EU.

### 2.3. Parasitological and Biopathological Analysis

Eggs per gram of faeces (FEC) was determined three times a week until slaughter. At the end of the study, the number of worms (male and female), the mean number of intrauterine eggs in female worms and the number of immature worms per gram of mucosa were assessed [22,23].

Blood and serum samples were collected once a week until the end of the experiment. Different hematological parameters, such as total and differential leukocyte counts, packed cell volume (PCV), plasmatic proteins and pepsinogen levels [24] were analysed in this study.

### 2.4. Determination of Systemic and Local Humoral (IgG and IgA Isotypes) Response

The antigen used to analyse the humoral responses was obtained from homogenates of adult *T. circumcincta* worms and subsequently purified by thiol-sepharose chromatog-raphy (TBSP), as previously described in the immunogen preparation. Systemic levels of specific IgA and IgG were determined from serum samples collected weekly throughout the study. At the end of study, mucus samples were also obtained by superficial scraping of the abomasal mucosa to analyse the local humoral response. The profiles of specific antibodies (IgG and IgA isotypes) against TBSP of adult worms in both types of biological sample (serum or gastric mucus), were performed by two indirect ELISA, as previously described [18,25].

### 2.5. Histology and Immunohistochemistry of Abomasal Mucosa

The number of eosinophils, globule leukocytes, and mast cells was determined in tissue sections stained with Giemsa and hematoxylin–eosin. Count was carried out in 40 randomly selected fields of 0.237 mm^2^, at the upper and lower third of the abomasal mucosa, and the results expressed as number of cells/mm^2^ [19].

For immunohistochemistry, samples were embedded in OCTM solution (Optimal Cutting Temperature, Tissue Tek, Sakura Finetek, Europe B.V. Zoeterwoude, The Nether-lands), followed by immersion in 2-methilbutane (Merk, Darmstadt, Germany) at −80 °C until processed. Sections (4 μm thick) from the abomasal mucosa were transferred to poly-l-lysine hydrobromide (Sigma-Aldrich Inc., USA)-covered slides. Primary monoclonal antibodies against CD4, CD8, CD45R, γδ, MHCII and WC-1 lymphocytes were diluted at 1:15, 1:15, 1:5, 1:10, 1:20 and 1:5, respectively. Sections were processed as previously described for avidin–biotin peroxidase (ABC) complex reaction, and immunolabelled cells were counted in 40 fields located in the upper and lower third of the mucosa [18].

### 2.6. Determination of Relative Cytokine Gene Expression by Real-Time PCR (RT-PCR)

Abomasal lymph node and abomasal mucosa samples were preserved in Trizol (TRI-Reagent, Sigma-Aldrich, USA) at −80 °C prior to RNA isolation [26]. cDNA synthesis was performed using 1 μg of total RNA and a reverse transcriptase kit following the manufacturer’s instructions (iScriptTM cDNA Synthesis Kit, BioRad Lab., Hercules, CA, USA).

RT-PCR was performed using the GoTaq^®^ qPCR Master Mix kit containing BRYT Green™ dye as fluorophore (Promega, Madison, WI, USA). For most assays a higher concentration of magnesium chloride (MgCl2) was used (Table 1). The amplification process was performed on an iCycler thermal cycler (BioRad, Hercules, CA, USA) fitted with a MyiQTM Single Color Real-Time PCR Detection System.

The sequence of the primers used for the cytokine gene expression (IL-2, IL-4, IL-10, INF-γ and IL17) and β-actine (reference gene) is depicted in Table 1. A relative quantification of gene expression was performed following the method ΔΔCt comparing Ct values obtained from samples of immunised or non-immunised and non-infected control animals. The data were normalised using the β-actine gene as reference gene assuming 100% efficiency of the assay [25,27].

**Table 1 vaccines-12-00437-t001:** Sequence of primers used in qPCR (accession number/reference), size (bp, in base pairs), melting temperature (Tm, in °C) of amplified products, and MgCl_2_ concentrations used in each reaction.

Cytokine	Primers 5′ to 3′	Size (bp)	MgCl_2_ (mM)	Tm (°C)	Accession Number/Ref.
B-act	CCAACCGTGAGAAGATGACCC CCCAGAGTCCATGACAATGCC	122	5	85.0	AF481159
IL-2	GTGAAGTCATTGCTGCTGGA TGTTCAGGTTTTTGCTTGGA	202	3	81.0	[28]
IL-4	GCTGGTCTGCTTACTGGTATG CGATGTGAGGATGTTCAGC	100	5	80.0	FJ936316
IL-10	GTGGAGCAGGTGAAGAGAGTC TGGGTCGGATTTCAGAGG	198	3	82.0	AF458378
INFγ	AGATAACCAGGTCATTCAAAGGAG GGCGACAGGTCATTCATCAC	180	3	82.5	U34232
IL-17	TGCTACTGCTTCTGAGTCTGGTGGC TGACCCTCACATGCTGTGGGAAGTT	111	0	83.5	[29]

### 2.7. Statistical Analysis

Data were statistically analysed using IBM SPSS Statistics for Windows, version 22.0 (IBM Corp., Armonk, NY, USA). The non-parametric Mann–Whitney U test was used to compare experimental groups on single-day parameters. To compare the data of the experimental groups on variables taken throughout the experiment (such as peripheral IgA and IgG levels, pepsinogen levels and packed cell volume), the general linear model for repeated measures was used, after testing normality of the data using the Shapiro–Wilk test. Variables that did not conform to normality were transformed using the square root function before applying the statistical model. Finally, the Spearman correlation test was used for the analysis of associations between the different parameters under investigation. Probabilities of *p* < 0.05 were considered significant.

## 3. Results

### 3.1. Parasitological Analysis

According to the results obtained in the coprological analyses (Figure 2A), the prepatent period was 14 days in the control group, while in the vaccinated group it was 19 days, showing throughout the experiment mean counts below those observed in the control group, particularly on day 30 p.i. when the faecal counts of the control group reached the highest mean value (800 epg). Consequently, the cumulative faecal counts showed lower mean values in the vaccinated animals, with a 21% reduction in cumulative counts at the end of the study; however, this reduction was not statistically significant.

At the end of the study, the mean value of adult worm counts in the vaccinated group was 1349 ± 567 worms/animal, with counts ranging from 778 to 2053 worms. On the other hand, the mean number of worms detected in the control group was 1760 ± 461 worms/animal, with a range of 1398 to 2416 worms/animal. According to this data, the immunised animals showed a 23.3% reduction in the total number of adult worms compared with the control group, although these differences were not statistically significant. Similarly, no statistical differences were observed when the data were analysed according to the sex of the adult worms. However, the reduction in worm numbers was higher in female than in male worms (32.4% vs. 12%) (Figure 2B).

Mean length of the female worms (mm) as well as the number of intrauterine eggs were similar between worms from both experimental groups. Thus, females only showed a 3.9% reduction in length (8.93 ± 0.7 mm vs. 9.30 ± 0.4 mm in the immunised and control groups, respectively). This reduction was also 8.8% in the mean number of intrauterine eggs of the worms from immunised animals (5.46 ± 2.2 vs. 6.03 ± 1.8). These differences were not statistically significant.

Moreover, a higher number of immature larvae were detected in the gastric mucosa of vaccinated animals in comparison to control group, with mean values of 10.3 ± 5.2 and 7.3 ± 5.2 larvae/gram of mucosa, respectively. However, these differences were not statistically significant. The analysis of statistical associations between the different parameters showed only a foreseeable significant positive association between the length and the number of intrauterine eggs in female worms.

### 3.2. Biopathological Analysis

All biopathological parameters analysed in this study (PCV, plasmatic protein and pepsinogen levels, as well as total and differential leukocyte counts) showed no significant differences between immunised and control groups. Analysis of the statistical associations between biopathological and parasitological data showed a negative association between the number of worms and plasma protein levels (r = −0.894; *p* < 0.05); however, this relationship was established independently of the immunisation status of the animal in both experimental groups.

When the different biopathological parameters were analysed (PCV, plasmatic protein and pepsinogen levels, and total and differential leukocyte counts), only a slight increase in serum pespsinogen levels was observed in the control group three weeks after experimental infection (p.i.), as well as an increase in peripheral eosinophils at 2 weeks p.i. in the immunised group. However, no statistically significant differences were detected between the groups. Concerning the statistical associations between biopathological and parasitological data, only a negative association between the number of worms and plasma protein levels (r = −0.894; *p* < 0.05) was detected, but this relationship was established independently of the immunisation status of the animals.

### 3.3. Systemic and Local Humoral Response

In vaccinated animals, the mean value of specific IgGs increased after the first immunisation. Subsequently, the mean level slightly decreased and increased again one week after challenge with L3 of the parasite, reaching the maximum level at 4 weeks p.i.—week 8 of the experiment—(72.12 ± 2.72 relative units), remaining elevated until the end of the experiment. Goat kids from the control group showed much lower anti-*T. circumcincta* IgG levels in serum than those observed in the immunised group, with a mean value of 14.95 ± 5.2 relative units, reaching a maximum value (24.63 ± 17.79) 2 weeks after the experimental infection (week 6 of the experiment).

Statistical analysis of the mean serum IgG levels over the course of the experiment using the general linear model for repeated measures showed significantly higher levels in the immunised group in week 2 of the experiment (during the immunisation), as well as from week 4 (challenge) to the end of the experiment, with *p*-values ranging from 0.012 to 0.042 (Figure 3A). When specific IgG levels peaked (week 8 of the experiment) or at the end of the study (week 10 of the experiment), negative associations were detected with all parasitological parameters analysed (except for the number of larvae in gastric mucosa), but in no case reached statistical significance.

The mean value of specific IgA observed in the vaccinated group was higher than that detected in the control group, though the differences were only significant 1 week after the second immunisation (week 3 of the experiment). At this point, levels of specific IgA showed (week 3) a noticeable negative association with parameters related to fertility of female worms such as cumulative FECs (r = −0.745, *p* = 0.013), length (r = −0.667, *p* = 0.05) and number of intrauterine eggs (r = −0.683, *p* = 0.042). After challenge, both experimental groups showed a slight increase in anti-*T. circumcincta* IgA in serum, but without statistical significance (Figure 3B).

Levels of specific antibodies at a local level (gastric mucus (IgGm and IgAm)) are depicted in Figure 4, where it is shown that the mean optical densities observed in the immunised group were higher than in the control group for both isotypes of immunoglobulins analysed, with significant differences for IgAm isotype (*p* = 0.050), but not for anti-*T. circumcincta* IgGm levels. Both specific IgAm and IgGm levels showed negative associations with the different parasitological parameters analysed, but these correlations were generally weak (with r-values ranging from −0.167 to −0.583) and did not reach statistical significance in any case.

### 3.4. Histology and Immunohistochemistry of Abomasal Mucosa

Mean count ± standard error of tissue populations of eosinophils, mast cells and globule leukocytes in the gastric mucosa from both experimental groups are shown in Figure 5. The gastric mucosa of the vaccinated animals showed higher eosinophil counts at the end of the study than that of non-immunised and infected goat kids (*p* = 0.048), generally accompanied by diffuse lymphocytic infiltration. This increase in tissue eosinophils was found to be negatively associated with adult worm counts at the end of the study (r = −0.872), with a *p*-value close to statistical significance (*p* = 0.054). Positive associations with specific IgA levels could also be found, especially at the local level, but without statistical significance.

No significant differences were detected between the two experimental groups in relation to the other two effector cells populations analysed (globule leukocytes and mast cells) (Figure 5A).

Lymphocyte subsets and antigen-presenting cells studied in the gastric mucosa (CD4, CD8, γδ, WC1, CDR45 lymphocytes and MHCII+, IgA+, IgG+ cells) showed higher average counts in the immunised animals than in controls, although significant differences were only detected when comparing, using a non-parametric test, the mean values of CD4 and CD8 lymphocytes as well as MCHII+, IgA+ and IgG+ cells from both experimental groups (Figure 5B).

### 3.5. Relative Cytokine Gene Expression by Real-Time PCR (RT-PCR)

In relation to the remaining cytokines analysed, the differences were less evident between both groups, showing, in some cases, a very low level of expression (IL-17). In contrast, the expression of the cytokines analysed in the mucosa of immunised animals was slightly downregulated compared with controls, particularly with regard to the expression of INFγ. This general trend did not hold for IL-4, whose expression was much higher in the mucosa of immunised animals, although without significant differences (Figure 6B). No clear associations were found between all these values and those obtained when analysing the different parasitological, histological and humoral parameters.

## 4. Discussion

This study attempted to evaluate the protective effect of thiol-sepharose chromatography-enriched fractions of *T. circumcincta*, similar to those used against *H. contortus* in small ruminants [15,30] and *O. ostertagi* in cattle [21]. These experiments were performed in goats over 9 months of age to ensure adequate development of the host immune response [31].

After analysing the different parasitological parameters, we observed a high variability in the results, in agreement with the results of other similar studies in this breed of goats [32]. Nevertheless, some trends were observed in the analysis of the abomasal content, where a 23.3% reduction in the total number of worms was recorded in the vaccinated group, with a more evident reduction in female worms (32.4%) than in male worms (12%). This was explained by the higher metabolic activity of females associated with reproduction, a pattern observed in other immunisation trials against *H. contortus* in sheep [33]. In female worms, a slight effect of immunisation on certain parameters associated with fecundity was observed, although without statistical significance, such as a reduction in the length of female worms and the number of intrauterine eggs.

These observations have been considered as two mechanisms of natural resistance to infection by these gastric nematodes in small ruminants [25,34] or as immunoprotective mechanisms in vaccination trials using thiol-sepharose chromatography-enriched extracts of *H. contortus* [35] or *O. ostertagi* [21,36].

All these observations could explain the reduction in the mean number of faecal eggs observed in the immunised group during the course of infection, which resulted in 21% lower cumulative counts compared with the control group. This immunoprotective effect, while improving the results obtained in sheep vaccinated with L4 antigens of *T. circumcincta* [37], is close to that observed in sheep of Canary Island breeds immunised with a vaccine based on eight recombinant antigens identified in *T. circumcincta* [38]. However, the reduction in FEC was much lower than in other studies using the above recombinant proteins, extracts of L4 *T. circumcincta* or adult *O. ostertagi* worms, which reported reductions in faecal counts of over 70%.

A more obvious immunoprotective effect has also been found in immunizations against *H. contortus*, including—for comparison with the present study—thiol-sepharose chromatography-enriched extracts, where the cumulative reduction in faecal eggs has been, in most cases, above 50%, and occasionally above 70% in both sheep [30,35] and goats [15,18].

The higher number of immature forms in the gastric mucosa of immunised animals, accompanied by a slight prolongation of the prepatent period, may be related to immunoprotection, similar to observations in experimental infections with *T. circumcincta* in sheep [38] or in cattle immunised with thiol-enriched antigens from *O. ostertagi* [11,21].

Biopathological parameters observed in both vaccinated and control groups were within the physiological ranges established for this species, probably due to the subclinical infection established by challenge [32]. As in sheep resistant to *T. circumcincta* [39,40], there was no peripheral eosinophilic response after challenge in our study, which has been observed in goats vaccinated with thiol-binding proteins of *H. contortus*, a response associated with immunoprotection [18].

In contrast to the biopathological results observed, the immunization carried out here generated an evident humoral response, at both systemic and local levels. The increase in specific antibody levels had already taken place during immunization, and triggered a secondary response after challenge, similar to that observed in goats vaccinated with similar antigens against *H. contortus* [15,31]. When comparing both immunisation assays at the local level, in both cases, there was an increase in specific antibodies in gastric mucus, a response in which the IgA isotype predominated against *T. circumcincta,* while against *H. contortus* it was the IgG isotype that showed higher levels [18].

The local response to *T. circumcincta,* observed here, has also been observed in sheep immunised with a recombinant prototype vaccine [41] and in sheep [42] and goats naturally resistant to the parasite [25]. In all these cases, specific IgA antibodies have been shown to be associated with defensive mechanisms against the parasite (mainly related to a reduction in fertility), supporting the idea of the relevance of this immunoglobulin isotype in immunoprotection against *T. circumcincta*.

Humoral response in vaccinated animals, after challenge, was consistent with the general idea that nematode infections produce Th2-type responses, and more clearly in previously sensitised animals, from which resistance to infection is established [27,43,44]. This humoral response also becomes settled after immunization with enriched fractions by thiol-sepharose chromatography against *O. ostertagi* [21], as well as *H. contortus* [18,19], a phenomenon that could be related to the higher gene expression of IL-4 (more than twofold relative to controls) observed in the gastric mucosa of the animals immunised in the present study, a cytokine characterised as Th2 [45].

However, at the level of abomasal lymph nodes, although gene expression was higher for all the cytokines studied, the most remarkable increase in the immunised animals corresponded to IL-2, a cytokine primarily involved in Th1 responses. Its function is directed towards macrophage activation and the induction of CD8 lymphocyte proliferation [46,47]. The increased IL-2 transcription in lymph nodes, which apparently seems to contrast with the results observed at the gastric mucosal level, could be related to the development of mixed Th1/Th2 observed by some authors in gastric nematode infections in ruminants after previous sensitization [41,48], or in cattle and sheep vaccinated against *O. ostertagi* and *T. circumcincta*, respectively [49,50]. Th17 and/or Treg-mediated responses may also play a role in generating protection against challenge [51]; however, both types of response did not exhibit a significant expression in the present study. In any case, all these results represent a preliminary insight into the responses generated in goats immunised against *T. circumcincta* with this type of antigen. This situation can also be influenced by numerous factors, including the specific parasite [28,46] and host species investigated in each case and the adjuvant used in the immunisation [52,53].

After analysing the results obtained in the immunohistochemical study of the abomasal mucosa, in the immunised animals, an increase in the mean count of all the cell populations studied was observed. This increase in cell populations is common in animals that show a certain degree of resistance to GIN infection [25,54], and these differences are significant for the MHCII+, CD4+, CD8+, IgA+ and IgG+ cell populations. Except for the CD8 lymphocyte subpopulation, these observations are in general in accordance with Th2 responses. The increase in antigen-presenting cells (MCHII) and CD4+ lymphocytes has also been found to be increased in the gastric mucosa of goats vaccinated with thiol-binding *H. contortus* antigens [18,19], as well as in animals with secondary GIN infections [54,55,56]. Concerning the increased number of CD8 lymphocytes in the gastric mucosa of immunised goats, although it could be related to the development of mixed Th1/Th2 responses considered above, it would be necessary to investigate further to rule out that the presence of these cells was related to mechanisms of evasion of the host immune response. The confirmation of this mechanism may open the possibility of using other adjuvants to improve the immunoprotection results obtained in the present study [57].

In addition to the increase in CD4+ cells in the gastric mucosa, another finding that suggests the activation of Th2 responses through immunization was the increased detection of IgA+ and IgG+ cells, which could have been related to specific antibodies in the mucus (especially of the IgA isotype in our study). This is a protective activity related to the reduction in the fertility of female worms and a recognised mechanism of resistance to *T. circumcincta* in small ruminants [57,58].

Th2 responses that could be generated by immunization with *T. circumcincta* thiol-binding fractions are accompanied by a significant eosinophilic infiltrate in gastric mucosa that is positively associated—although without statistical significance—with local levels of specific IgA. The increase in the number of eosinophils in the vaccinated animals could be contributing to the levels of IL-4 observed in the gastric mucosa of this group [45], also determining an immunoprotective effect very close to statistical significance in relation to the number of worms observed after challenge, as reported in sheep resistant to *H. contortus* [43,59].

Other common findings in animals resistant to GIN infections is the infiltration of mast cells and globule leukocytes [60]; however, data obtained in the current study showed no relevant contribution of both effector cells to the immunoprotection against *T. circumcincta*.

As previously mentioned, when comparing the degree of immunoprotection obtained in the present study in relation to other immunization trials in this same goat breed, in which thiol-binding fractions of *H. contortus* were also administered in complete and incomplete Freund’s adjuvant, a high protection against the parasite was observed, with a reduction in the number of worms of 65.5% and in cumulative faecal counts of 73.2%. Trying to analyse whether these differences were related to the immune responses generated in both cases, a humoral response was observed at both local and systemic level, being more relevant to the IgA levels in the gastric mucosa in the immunisation against *T. circumcincta*. In both cases, eosinophil levels appeared increased after challenge in immunised animals, establishing negative associations with parasitological parameters, while no relevant activity was detected in other effector cells. In the gastric mucosa, animals vaccinated with this type of enriched (thiol-binding) extract showed a cellular infiltrate with increased mean levels of MCH-II and CD4+ cells. In contrast, the most significant differences were found in the levels of CD8+ lymphocytes, which were proportionally higher in the group of animals immunised against *T. circumcincta* [18].

In summary, the responses generated after immunisation with thiol-binding enriched fractions in goats from adult worms of both gastric nematodes (*T. circumcincta* and *H. contortus*) were very similar—except for some differences referred to the CD8 lymphocyte subpopulation—which seems to indicate a higher efficacy of the defensive mechanisms generated by this type of immunisation when they take place against *H. contortus*, which could be justified by the neutralizing effect of antibodies to parasite components at the intestinal level, which is much more effective in haematophagous parasites such as *H. contortus* when ingesting host blood, a mechanism proposed to explain the protection conferred by immunisation with this type of immunogen [18,61].

## 5. Conclusions

The current study shows that thiol-binding enriched fractions from *T. circumcincta* induced a partial protective effect against the parasite in goats, resulting in a slight reduction in the number of worms (23.3%), mainly affecting female worms, which also showed a slight mean reduction in their length and in the number of intrauterine eggs, as well as a 21% reduction in cumulative faecal eggs. The immunoprotection levels of this type of enriched fraction were lower than those observed in goats against *H. contortus*, but generated similar Th2-related immune responses, showing a prominent humoral response (especially of the IgA isotype at local level) and eosinophil infiltrate in the gastric mucosa (which are negatively associated with the fertility of female worms and the number of worms, respectively), as well as an infiltrate of MCHII antigen-presenting cells and CD4+ lymphocytes, IgG+ and IgA+ cells. In addition to these responses, and in contrast to the results obtained in previous immunisation trials against *H. contortus*, a significant infiltrate of CD8+ lymphocytes was detected in the gastric mucosa, as well as an increased gene expression of IL-2 in abomasal lymph nodes. The role of CD8+ and IL-2 may be associated with inhibitory activity of the protective responses induced by immunisation or linked to the development of mixed Th1/Th2 responses. This aspect should be subject to future analysis to enhance the level of protection, particularly through the use of adjuvants and/or immunomodulators.

## Figures and Tables

**Figure 1 vaccines-12-00437-f001:**
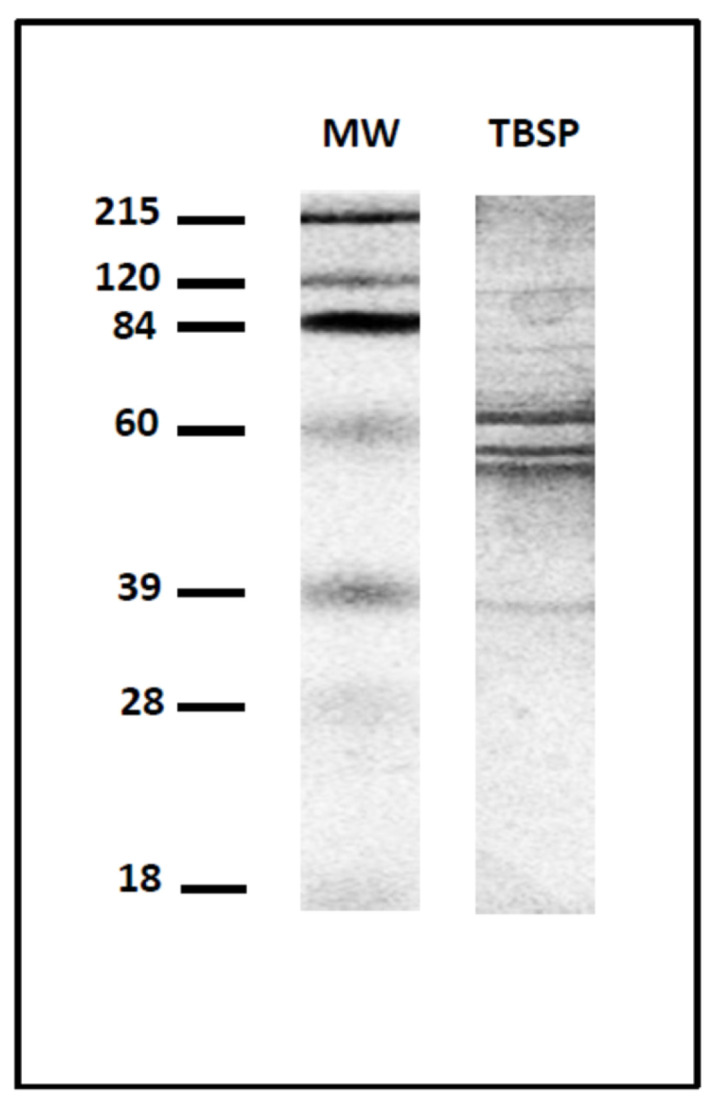
SDS-PAGE electrophoresis under non-reducing conditions of the antigenic extract used for the vaccination and the analysis of the humoral response. TBSPs: thiol-binding somatic proteins from *T. circumcincta* adult worms; MW: molecular weight marker in kDa.

**Figure 2 vaccines-12-00437-f002:**
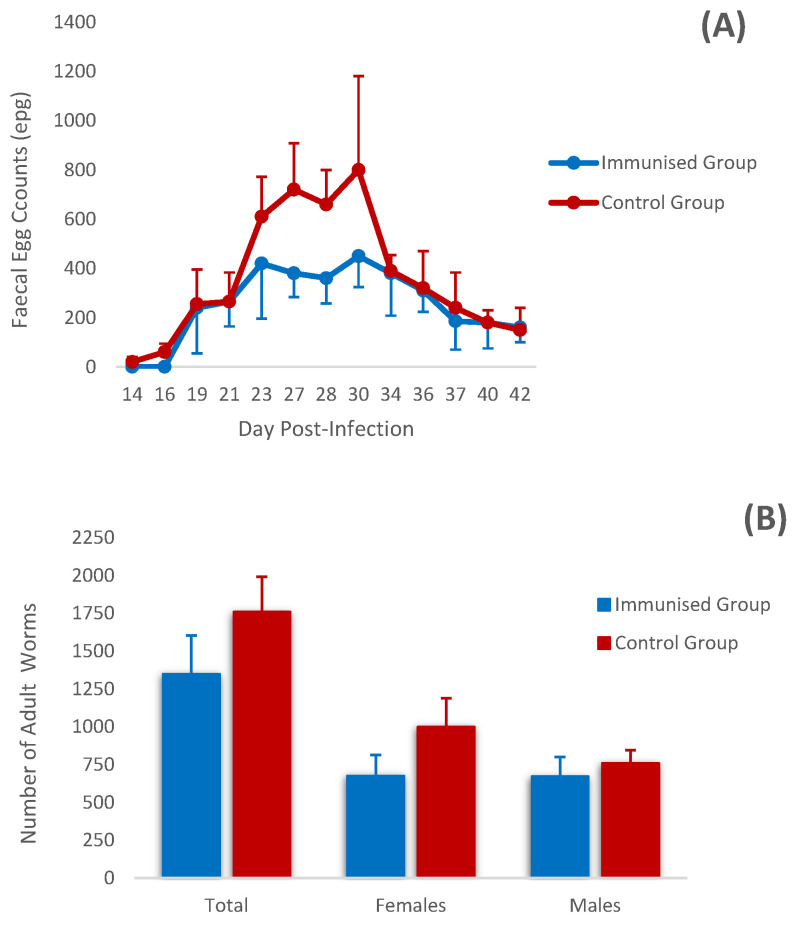
(**A**) Faecal egg counts in immunised and control groups after challenge. Results are expressed as mean values of eggs per gram (epg) of feces. Results are mean ± S.E.M. (**B**) Mean adult worm counts (total, females and male worms) in goats from the immunised and control groups at the end of the experiment. Results are mean ± S.E.M.

**Figure 3 vaccines-12-00437-f003:**
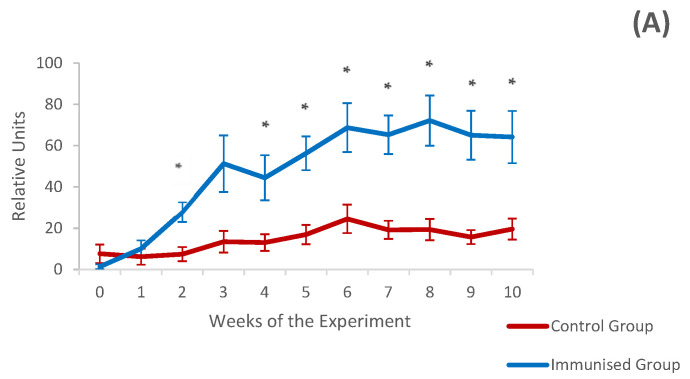
Evolution of levels of specific IgGs (**A**) and IgA (**B**) in serum against thiol-binding somatic protein (TBSP) fractions from *T. circumcincta* in goats from the immunised and control groups throughout the experiment. Results are mean ± S.E.M. * *p* < 0.05.

**Figure 4 vaccines-12-00437-f004:**
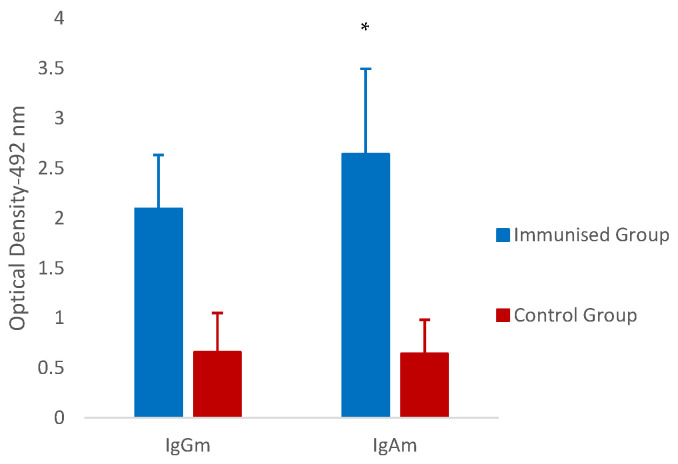
Mean mucosal IgG (IgGm) and IgA (IgAm) against thiol-binding somatic protein (TBSP) fractions from *T. circumcincta* in immunised, and control goats at the end of the experiment. Results are mean optical density at 492 nm ± SEM. * *p* < 0.05.

**Figure 5 vaccines-12-00437-f005:**
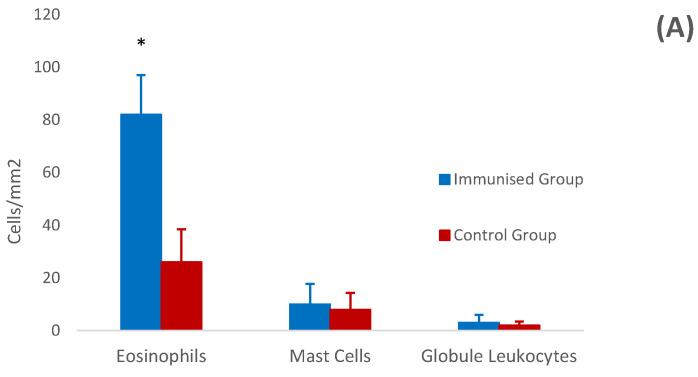
Level of effector cells (**A**) and cellular subpopulations (**B**) in the gastric mucosa in goats from immunised and control groups. Results are mean number of cells/mm^2^ ± SEM. * *p* < 0.05.

**Figure 6 vaccines-12-00437-f006:**
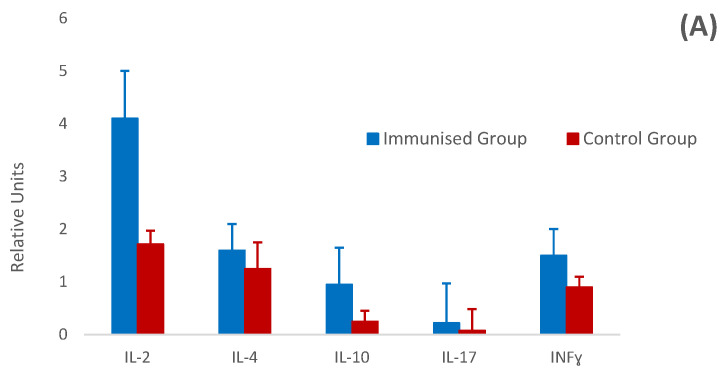
Gene transcription levels of IL-2, IL-4, IL-10, IL-17 and INF-γ in abomasal lymph nodes (**A**) and gastric mucosa (**B**) from immunised and control groups. Results are mean relative units (RU) ± SEM after applying the ΔΔCt method (giving a value of 1 RU to the mean value observed in the non-immunised and non-infected group) and using β-actin as reference gene.

## Data Availability

The data presented in this study are available in this article.

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
