# Peer review of "Analysis of Protection and Immune Response against Teladorsagia circumcincta in Goats Immunised with Thiol-Binding Proteins from Adult Worms"

_vaccines, 2024, doi:10.3390/vaccines12040437_

Round 1

Reviewer 1 Report

Comments and Suggestions for Authors

The study entitled "Analysis of protection and immune response against Telador- 2 sagia circumcincta in goats immunized with thiol-binding protein- 3 teins from adult worms" presents new evidence for a promising field in veterinary parasitology. I have brief comments on the manuscript.

Please review the standard template used to prepare the summary. The authors even seem to include similar topics in the discussion.

Did the animals undergo treatment with anthelmintics prior to the start of the study? If yes, report on the time interval between medication administration and the start of the study.

Was the animal infection monospecific?

In conclusion, the authors state that there was a "slight reduction" in the number of worms and the number of eggs. But they show that there was a statistical difference between the treated group and the control group. Please review these statements.

The authors need to include results from this study only in the conclusion. Possible comparisons with previous studies are more recommended to be made in the discussion.

Reviewer 2 Report

Comments and Suggestions for Authors

This manuscript described the analysis on the protection and immune response against Teladorsagia circumcincta in goats immunized with thiol-binding proteins from adult worms. As there are very few reports studying vaccines against parasitic nematodes infecting goats especially for T. circumcincta although one study reporting vaccination experiments in sheep using recombinant proteins, the findings from this study enriched our understanding of vaccine potential of thiol-binding proteins of T. circumcincta. The aim of this study was well formulated, the experimental methods were reasonably selected, the analyses were accurately performed and the results were clearly presented. I would suggest to accept for publication after the following revision.

1.      Please check typographical errors throughout the manuscript. For example, on page 240, “temperatura” should be changed into “temperature”. On page 249, “packet cell volume” should be changed into “packed cell volume”. On pages 107, 108, 118, 350-351, “T. circumcincta” should be italic.

2.      Figures 1 and 2 can be combined into one figure, the same to Figures 3 and 4. In addition, Figures 5 and 6 can be combined into one figure.

3.      The discussion section is too long, please make it more concise.

Comments on the Quality of English Language

There are some typographical errors which need to be checked and corrected throughout the manuscript. 

Reviewer 3 Report

Comments and Suggestions for Authors

Manuscript vaccines-2923098 was reviewed. The manuscript describes a vaccination trail with thiol-binding proteins against T. circumcincta. It is an interesting topic in the light of the development of drug resistance. The MS is generally well written, except it contains some very long, and therefore confusing sentences.

My main comment involves the antigens.

1) Although the procedure of isolation of the vaccine is well described in Material and Methods, the results of these complicated purification is not given. Now we have no idea if there is any enrichment of the thiol-binding proteins in the vaccine compared to the crude extract. Some SDS-PAGE gels of the purified proteins compared with the crude extract are highly recommended.

2) It seems to me that the antigen used for ELISA is different from the antigens constituting the vaccine (line 140-150). Why was the ELISA not performed with the same antigens? Although in the legends of figure 3 was stated that the anti-TBSP was measured, from lines 145-150 it seems that a more crude extract was used. Figures 4 and 5 stated only “against T. circumcincta”. Were these different antigens? A positive correlation between anti-TSBP and protection within the vaccinated group (when present) would have been a strong indication that TSBP can be protective. Furthermore, already after vaccination, but before infection a correlation with protection can possibly be found when the same antigens from the vaccine were used as antigen in ELISA. A better description of the used antigen is highly recommended.

Minor points:

Line 36. At first use Teladorsagia circumcincta, thereafter, T. circumcincta. Also for the other species.

Line 107. Here and elsewhere: T. circumcincta is not in Italic.

Line 265. A figure of the EPGs in addition to or instead of the cumulative EPGs can be informative.

Line 295. Add “numbers of” after “higher”

Line 324. Change “slight” into “slightly”.

Line 325. When the challenge infection was at week 5, 4 weeks pi will be week 9, not 8.

Line 330. 2 weeks after pi will be week 7, see also line 325.

Line 333-336. The week numbers are mixed up. Why not stick to the weeks of the experiment throughout the manuscript, in order to avoid confusion.

Line 434. Change into O. ostertagi.

Line 464-468 and Line 475-478 and line 488-492 and line line 492-496 and line 509-518! And many more. Very complicated and long sentences. Rephrase.

The Discussion is very long. Can be more to the point at places

Comments on the Quality of English Language

Line 464-468 and Line 475-478 and line 488-492 and line line 492-496 and line 509-518! And many others are very complicated and long sentences. Rephrase.

Round 2

Reviewer 3 Report

Comments and Suggestions for Authors

welll performed experiments and well written paper. No further comments

Author Response

Dear Reviwer #3,

Thank you very much for your review and suggestions on the first version of our article, which have undoubtedly helped to improve it.

Best wishes